# Mental Health Problems among Young People—A Scoping Review of Help-Seeking

**DOI:** 10.3390/ijerph19031430

**Published:** 2022-01-27

**Authors:** Katrin Häggström Westberg, Maria Nyholm, Jens M. Nygren, Petra Svedberg

**Affiliations:** 1School of Health and Welfare, Halmstad University, SE-301 18 Halmstad, Sweden; maria.nyholm@hh.se (M.N.); Jens.nygren@hh.se (J.M.N.); petra.svedberg@hh.se (P.S.); 2Affecta Psychiatric Clinic, Sperlingsgatan 5, SE-302 48 Halmstad, Sweden

**Keywords:** mental health, young people, help-seeking, scoping review, user perspective, qualitative

## Abstract

Young people’s mental health is a public health priority, particularly as mental health problems in this group seem to be increasing. Even in countries with supposedly good access to healthcare, few young people seek support for mental health problems. The aim of this study was twofold, firstly to map the published literature on young people’s experiences of seeking help for mental health problems and secondly to validate whether the Lost in Space model was adaptable as a theoretical model of the help-seeking process described in the included articles in this scoping review. A scoping review was conducted in which we searched for literature on mental health help-seeking with a user perspective published between 2010 and 2020 in different databases. From the 2905 studies identified, we selected 12 articles for inclusion. The review showed how young people experience unfamiliarity and insecurity with regard to issues related to mental health and help-seeking. A strong wish for self-reliance and to safe-guard one’s own health were consistent among young people. Support structures were often regarded as inaccessible and unresponsive. There was a high level of conformity between the model on help-seeking and the analysed articles, reinforcing that help-seeking is a dynamic and psychosocial process.

## 1. Introduction

Young people’s mental health is a major public health issue. Mental health problems among young people contribute to impaired physical and mental health extending into adulthood [1,2,3]. Promoting young people’s mental health is an integral component in ensuring their development and improving health and social wellbeing across their lifespan [3]. In light of the high rate of mental health problems among this group, a corresponding high rate of help-seeking and use of support resources might be assumed; however, few young people actually seek and eventually access professional help. Delays in looking for help can be lengthy and are prevalent even in countries with good access to healthcare [4,5,6,7,8,9]. The process of searching for support involves barriers that relate to both individual and social context factors [8,10]. This contributes to the complexity involved in offering interventions to support them and highlights the need to understand the help-seeking process, whether online or in person, for young people with mental health problems.

Help-seeking is usually described as a rational, agency-based process where the individual plans, decides and acts on symptoms [11]. However, research also describes that help-seeking is not solely an individual act; rather, it is influenced by social factors throughout the process. Societal, organizational support structures set the limits and stipulate the opportunities to seek help [12]. Help-seeking thus depends both on factors at the individual level and structural resources for young people. Many studies that examine help-seeking for mental health among young people using cross-sectional designs on either the general community, or school populations [8,13] are based on descriptive data that is often generated through surveys, and focus on attitudes, rather than on experiences [14]. The main focus of previous literature has been on individual factors, such as mental health literacy, and less information can be found on the structural factors involved [8,13]. This calls for a deeper and more nuanced understanding of young people’s mental health help-seeking regarding contextual factors, with particular focus on their experiences and perspectives. An improved understanding of help-seeking for mental health problems can be used to improve practice and service delivery, and ultimately benefit young people’s mental health.

In this study, qualitative research exploring the help-seeking process in Sweden from the perspectives of young people with mental health problems was used as the theoretical point of departure [15]. Within this previous research, we produced a theoretical model of help-seeking, the Lost in Space model [15]. It showed how help-seeking was a long, non-sequential and dynamic process. In this research, young people described a process of moving in and out of the three help-seeking phases, Drifting, Navigating and Docking. Drifting was characterized by insecurity and unfamiliarity, with a lack of knowledge of mental health and the support system; Navigating was characterized by structural obstacles, a fragmented support system and wishes for help; while Docking was characterized by experiences of finding help. For the purpose of confirmability and usefulness, it is essential to validate and understand if the model can be applied to other settings and contexts—for example, whether the model is consistent with the experiences of help-seeking by young people in other countries. Therefore, the aim of this study was twofold, firstly to map the published literature on young people’s experiences of seeking help for mental health problems and secondly to validate whether the previously published Lost in Space model was adaptable as a theoretical model of the help-seeking process described in the included articles in this scoping review.

## 2. Methods

A scoping review was deemed the most preferable approach to responding to this broad area of interest [16]. Scoping reviews maintain a broad window for inclusion of studies of a range of types and levels of quality [17]. Our scoping review protocol was developed using the scoping review methodological framework proposed by Arksey and and O’Malley, entailing five framework stages. The framework was further developed by Levac, with a qualitative elaboration of the material [17,18]. These stages provide a clear sequential order in which to identify and collect studies, chart the data and report results, and the scoping review protocol was used for guiding the research.

### 2.1. Stage 1: Identifying the Research Question

A multidisciplinary research team with experience of health science research, including public health, nursing, and youth research was assembled to discuss and clarify the scope of inquiry and identify research questions. The target population of interest was defined as young people (ages 11–25) with experience of mental health problems, and experience of help-seeking in that regard. Mental health problems were defined as commonly experienced problems of depression or anxiety, as well as behavioural and emotional problems. Considering the concept of help-seeking, the term is used to understand the delay of care and to explore possible pathways for mental health promotion. For this study, help-seeking was defined as seeking and/or accessing professional help for mental health problems. Conceptually, help-seeking was regarded as a process influenced by social, psychological and contextual factors [12]. The research questions for this study were (1) to map general characteristics of published literature focusing on the young people’s experiences of seeking help for mental health problems, and (2) to explore how the previously published theoretical model Lost in Space could be further refined and complemented via an abductive approach, drawing the final set of categories and themes informed by the papers reviewed in this study.

### 2.2. Stage 2: Identifying Relevant Studies

A search strategy was developed in collaboration with a librarian to develop search terms using subject heading terms adapted to each of the three included databases: Medline/PubMed, PsycINFO and CINAHL. The search terms for the target population were adolescents, young and emerging adults; for the health outcome, they were mental health, depression, anxiety, and for the concept of interest, the term was help-seeking. Other criteria were limiting searches to studies written in English, and studies being published between 2010 and 2020 due to rapidly evolving research and policy changes in this area as well as the increased rates of mental health problems among young people. The searches were conducted during summer 2020. See Appendix A for the full search strategy.

#### Inclusion and Exclusion Criteria

Studies were eligible for inclusion if they investigated help-seeking among young people with mental health problems aged between 11 and 25. Only studies that specifically investigated young people’s own perspectives of experiencing or having experienced mental health problems and help-seeking were included. Since the intention was to understand help-seeking among young people with common mental health problems, studies on particular target groups or populations were excluded, such as studies on specific treatment interventions. Likewise, studies focusing on help-seeking attitudes or potential help-seeking intentions of general populations without personal experience of mental health help-seeking were excluded. Studies had to specifically focus on adolescents or young people; thus, studies with a more population-based perspective, or encompassing wider age groups, were excluded. Theses were not included as it was assumed that any material within a thesis on help-seeking, that otherwise fitted the inclusion criteria, would appear as published articles. Comments, editorials, consensus statements and other opinion-based papers were excluded, along with studies solely exploring the perspectives of others, other than the help-seekers themselves (e.g., families, helpers, professionals, etc.).

### 2.3. Stage 3: Study Selection

All identified studies from the searches were imported to the management reference tool EndNote, version 20.1, and duplicates were removed. Screening was carried out with a sequential, stepped approach and an iterative process between the authors of the study [18]. In the first step of study selection, titles and periodically abstracts were screened by KHW, who discarded obviously irrelevant studies based on the exclusion criteria. In the second step of study selection, abstracts of the remaining studies were screened independently by three of the authors (KHW, PS and MN) to determine eligibility based on the defined inclusion and exclusion criteria. Disagreements between the authors were discussed with a fourth author (JN) until consensus was reached. The third step required KHW to examine the full-text of the remaining articles to determine eligibility, subsequently discussing the articles with all authors. A PRISMA diagram (Figure 1) details the screening process with number of papers retrieved and selection of the included studies.

### 2.4. Stage 4: Charting the Data

Data charting was conducted in accordance with scoping review standards using a template that was developed for the extraction of information from each study regarding the following: authorship, year of publication, journal, source of origin, design, population and age group, aims of the study, methodology and important results [17]. A descriptive, numerical summarization was made, presenting the extent, nature and scope of included studies [18], see Appendix A for the full bibliographic information of the included studies.

### 2.5. Stage 5: Collating, Summarizing and Reporting Results

A qualitative thematic analysis was conducted to examine and aggregate the findings from the help-seeking process, as depicted in the included studies [18]. For the thematic analysis, an abductive approach was taken [19], based on the previous Lost in Space model [15]. According to such an approach, hypotheses can be explicated through deduction and verified through induction. Abduction thus means that new explanations are based on background theories and, whilst taking empirical material and restrictions into account, may lead to elaborated knowledge [19].

The analysis began with reading the findings in the included articles several times, then identifying and inductively coding text and quotes [20] in relation to young people’s experiences of seeking help for mental health problems. In this phase, data were inductively scrutinized to discover experiences, expressions and perspectives, keeping codes close to the data; for example, the text ‘Some young people reported that discussing uncomfortable emotions was unfamiliar’ was coded as the theme Unfamiliarity. The deductive process followed, in which the theoretical model Lost in Space was employed. It describes help-seeking among young people with mental health problems in a Swedish context [15]. A categorization matrix was developed based on the model, emanating from the original subcategories and categories, the themes within the subcategories and the properties of themes. The deductive process in the analysis involved going back to the data and placing the inductively derived codes into themes and subcategories of the theoretical model. All themes from the original model were found through coding the analysed articles. Codes from the new material that did not match the original theoretical model subcategories contributed with new aspects to existing themes of the model and, in some cases, generated new themes, thereby broadening the understanding of help-seeking. In one instance, the name of one subcategory was altered to reflect new material. KWH performed the data analysis and, to enhance the quality and validity of the analysis, the data analysis was discussed continuously with all authors.

## 3. Results

### 3.1. Mapping the Characteristics of Published Literature

In total, 1540 articles were identified as potentially relevant records, after duplicates were removed through the database searches. After the first screening of title and abstract, 1207 articles were excluded on the basis of age, format type, content (i.e., not dealing with help-seeking), focusing on specific populations or not being based on a user perspective. In the second round of screening, another 243 articles were excluded due to the eligibility criteria. In the third round of screening, the remaining 90 articles were reviewed in full-text and of these 12 articles met the full set of eligibility criteria.

The characteristics of the included studies are described in Appendix A. Seven articles were published between 2010 and 2015, and five after 2016. The designs were mostly qualitative, with individual interviews (*n* = 9) and focus groups (*n* = 7). Seven articles employed a combination of methods (for example, mixed methods), and two articles included information from surveys. The focus of articles covered: social and organizational factors impacting help-seeking, functional concerns, attitudes towards computerized mental health support, attitudes to consulting primary care, perceptions and help-seeking behaviours in schools, exploration and identification of barriers and facilitators in general populations with and without previous experience of mental health support, barriers and facilitators in male groups, links between masculinity and help-seeking, comparisons of groups’ help-seeking strategies and descriptions of experiences, self-management and help-seeking. The recruitment of participants varied, utilising educational settings (*n* = 4), youth mental health services (*n* = 2), community websites (*n* = 1), primary care (*n* = 1), youth services (*n* = 2), previous participation in longitudinal studies (*n* = 2) and community samples (*n* = 3). Four articles focused specifically on young males, and four on barriers to help-seeking. Three articles were set in the USA, one in Canada, three in Australia and five in Europe. The age range, 11–25, was seen in a variation of age clusters, with the smallest age range being two years (ages 20–22) and the largest 13 years (ages 12–25); the mean age range covered was six years. 

### 3.2. Examination of the Help-Seeking Process from the Perspectives of Young People

The findings from this examination showed a high level of agreement with the theoretical model Lost in Space. Overall, the results showed that help-seeking was a dynamic and psychosocial process without sequentially fixed stages, where young people expressed an unfamiliarity with, insecurity about and lack of knowledge of mental health issues, a longing for self-reliance and, in some contexts, a presence of stigma. Young people did not consider the support structures to be responsive or accessible. Below, Figure 2 outlines the examination of the help-seeking process from the perspectives of young people. It includes confirmed content of the old model, new content derived from the analysed articles, and elaborations according to the abductive method. The ‘number of meaning units’ refers to coded material in the analysed articles. ‘Original’ refers to subcategories and themes from the Lost in Space model, where findings were corroborated by codes from the analysed articles (‘confirmed content’), other elements that emerged showed further dimensions of experiences that contributed to new perspectives of established subcategories in the model (‘new content’), and some themes that emerged in the analysis were not readily encompassed within the subcategories in the original model (‘new’) (see Figure 2).

#### 3.2.1. Drifting

Drifting, the initial category of the Lost in Space model, encompassed a general feeling of unfamiliarity, lack of knowledge, trivialising oneself and problems due to insecurity often by normalizing and minimizing one’s experiences. Young people’s voices in the analysed articles corroborated Drifting well, through similar expressions and experiences. 

##### Fumbling in Life

In the original model, Fumbling in life encompassed themes of unfamiliarity, insecurity and trivialisation. Likewise, young people in the analysed articles expressed unfamiliarity with both mental health problems and not recognizing oneself. Lack of knowledge was frequently described both with regard to communicating, distinguishing and assessing emotions but also regarding where and when to seek help, leading to a sense of insecurity. Because of this insecurity, young people practised trivialisation, trying to make their problems smaller or unimportant. They also had the impression and fear that their problems would not be sufficient to receive support. In some of the analysed articles, trivialisation was presented as a rational strategy, enabling young people to downplay their problems and rationalizing not actively dealing with them or approaching others for help, whereas, in the original model, trivialisation was carried out due to a sense of insecurity. In this section, no conceptual changes to the original model are suggested.

##### Struggling

In the original model, Struggling was characterized by simultaneous descriptions of mental health problems and incessant attempts and strategies to feel better, ambivalence and a longing for self-reliance. These themes re-appeared in the included articles. Mental health problems were described by the participants as emotional problems, panic attacks, sadness, self-harm, anxiety and lack of motivation. Within Struggling in the original model, young people usually referred to mental health problems as being something “within” (internal) rather than originating “outside” (global). However, in several of the analysed articles, the mental health problems were attributed to something “outside”. Hence, young people also related mental health problems to relationships, stress and risk-taking behaviour.

Themes on endeavouring strategies trying to deal with mental health problems were common in the included studies, as was also the case in the Lost in Space model, pointing to the more-or-less continual and relentless attempts and strategies young people performed in order to deal with their problems. Although it was proposed by young people in individual studies that seeking help requires effort, lack of effort was not a dominant issue for young people in either the original model or in most analysed articles. In some articles, an in-depth exploration of the strategies employed was undertaken, according to having an ‘approach’ or ‘avoidant’ character, or gendered differences, adding to the variation in strategies, whereas in the model, an abundance of strategies was ascertained; however, the type of strategy was not explored. Denial was a common strategy in both the original model and in the included articles. In the Lost in Space model, this was described as “shutting off”, with the intent of ignoring feelings and problems. This strategy was directed towards oneself: wanting to manage things, being strong and coping. In some of the analysed articles, denial was presented as relating to a sense of embarrassment, or as being done in order to protect others. Several reasons were attributed to this phenomenon: that young people did not want to trouble others, did not want to burden or alarm others, and did not trust others. In the model, reasons for denial were differentiated by a sense of responsibility, enacted by, for example, not sharing information with family and friends. Withholding information thus seemed to relate to aspects additional to a sense of responsibility and self-reliance.

A frequent theme in both the original model and in the findings from the included articles was self-reliance. Statements of wanting to be strong, trying to cope on one’s own, not sharing information and an elevated sense of responsibility to manage one’s life and mental health problems were evident. Ambivalence as a theme recurred throughout the material, in both the included studies and the original model. Young people expressed simultaneous and contradictory feelings and thoughts towards both themselves and their problems, others and help-seeking per se. They were often hesitant to seek help, whilst at the same time expressing a need and a longing for help.

The analysis of the included articles suggested no major conceptual changes to the original model, although the themes Endavouring strategies and Mental health problems are both elaborated. 

##### Reaching a Point of No Return

Within Reaching a point of no return in the original model, young people expressed deterioration and a reaching out for support, often with the help of others. In Lost in Space, others were called ’catalysts’, showcasing their importance in actually initiating a help-seeking process. Within the analysed articles, important others were consistently brought up by the young people, with examples of others coaching, supporting, guiding and, in some instances, taking control of the help-seeking process. A new perspective in the included articles was a negative perception of control, and how others exerted control over them, compelling them to seek help. While this aspect of negative control did not emerge in the original model, an elaboration of the model may expand on the various functions of the important others, e.g., by dividing them into controllers vs. supporters. The other theme in this subcategory, deterioration, was brought up in several articles, as in the Lost in Space model. This indicated a worsening of symptoms and a decreased ability to function. Young people described not leaving the house, escalated behavioural problems, self-harm and suicide attempts, or ‘having a melt-down’ as triggers for seeking help. Young people also described how their problems were ‘revealed’ and others became aware of their problems, which in turn led to seeking help. 

The included articles emphasise that seeking help is often a long process that takes place during a prolonged time-span. Therefore, in this section, a change of title of the subcategory Reaching a point of no return, to Transitioning towards decision, is suggested. 

#### 3.2.2. Navigating

The category Navigating depicted attempts of trying to find support, personal reflections, hopes and longings and wrestling with structural barriers. Expressions from young people in the analysed articles conformed well with the subcategories Trying to dock and Wrestling with structure.

##### Trying to Dock

This subcategory in the Lost in Space model entailed descriptions of personal reflections, hopes, longings and disappointment when trying to seek support. All themes from the original model were exemplified in the included articles. Hopes for help, as well as being safe, noticed and understood, were common in the included articles, as were accounts of the opposite, feeling unsupported. Miscommunication while not being understood or listened to also appeared in both materials as did accounts of being treated like a child and not taken seriously, thus containing references to issues of power. Several analysed articles contained descriptions by young people on how support was perceived as impersonal and instrumental rather than person-centred. This added aspects of negative references to professionalism and reliance on medication. Young people expressed the importance of reframing negative and medical terminology in positive and informal terms. Both materials contained descriptions of young people feeling unsupported, which led to continued and continual efforts of seeking support. A new theme, trust, was identified in the thematic analysis from descriptions of lack of confidence in treatment, and how familiarity facilitated help-seeking. A lack of trust was depicted as arising from limited prior contact, from anxiety about seeking help, from concerns about professional competence and from negative perceptions of professionals. Within the theme of trust in the articles, concerns about confidentiality and parental involvement surfaced, whereas, in the original model, these concerns were interpreted as structural obstacles.

Common themes in the articles were stigma and shame, whereas in the original model, this was not pervasive. The included articles relayed young people’s strong sense of shame about seeking help. They perceived it as a display of weakness. Fear of social consequences, ridicule and a longing to fit in led young people to describe a feeling of shame or embarrassment, and to having thoughts of what others would think and say. They also made efforts to conceal both mental health problems and help-seeking. Articles focusing exclusively on males stressed the gendered aspect of this, claiming that this group was affected by masculine ideals of strength and autonomy, which hindered displays of weakness and prevented help-seeking. In the original model, some findings relating to this theme were described; however, the term stigma was never used. Instead, this was described in the subcategory Wrestling with structure, in relation to seeking support in school, with references of embarrassment and an undesirable show of weakness in front of peers. 

In this section, the analysed articles provide more aspects on the Feeling unsupported and Miscommunication themes. The large presence of codes in the new material relating to Stigma and Trust suggests the incorporation of Stigma and Trust as unique themes into the model.

##### Wrestling with Structure

In both the original model and the analysed articles, there were multiple references to structural obstacles, such as access, waiting times, resources, continuity, inadequate chains of support, and lack of coordination between supporters. Young people voiced feelings of not being met by professionals in an appropriate and timely manner, and concerns about how they were passed on, being referred to other support structures, and how there was a perceived lack of resources, making access difficult. Help-seeking was described as inconsistent, with repeated attempts at initiating and discontinuing help. Young people in several articles, and the original model, expressed that primary care was not an option when seeking support. Primary care was regarded as handling physical health complaints and that its practitioners were not being skilled in mental health issues. Particularly for the ‘younger’ of the young people, expressions that primary care was not directed at their age group were voiced. The inadequate support services theme was thus corroborated by young people in other contexts.

Confidentiality and age issues were concerns for the young people, both within the original model and the thematic analysis, primarily relating to parental control and insight. Both materials contained descriptions of how young people assumed and were concerned that confidential information shared with professional supporters would be communicated to parents. In some articles, this was said to relate to the theme of trust; however, confidentiality was mainly related to being a minor lacking power. Young people also voiced that being a minor was as an obstacle for independently accessing help. Likewise, age was an issue for the ‘older’ young people, who reported feeling out-of-place at youth-specific services. In the original model, a sense of resignation, often related to difficulties accessing support and feeling unsupported, was evident. The included articles provided additional material relating to this, as a sense of powerlessness appeared in several subcategories, and in the process as a whole.

The analysis supported a clearer conceptual division between subcategories Wrestling with structure and Trying to dock in the model. The latter entailed primarily personal accounts and experiences, expressions of hopes, disappointments and recounts of feelings, and the former referred primarily to structural conditions. Recurring references in relation to powerlessness suggest this is elevated to a permeating theme, capturing young people’s experience of seeking help.

#### 3.2.3. Docking

Docking in the original model contained references from young people to the subcategories Finding support and Changing as a person.

##### Finding Support

All original themes of the subcategory Finding support were found in the thematic analysis. In both the original model and the analysed articles, young people described experiences of being validated, accepted, recognized and listened to. The importance of the comfort of support and initial positive contact was stressed. Descriptions of good and bad supporters and preferences regarding, for example, gender and profession, were evident. Both materials contained descriptions of negative outcomes and unwanted consequences from having sought help; for example, in the original model, this was described as problems being exaggerated and social services becoming involved. In the analysed articles were descriptions of referrals to support services appearing as punitive rather than helpful. This subcategory also contained accounts in both the analysed articles and the original model of young people being disregarded and not being taken seriously. 

In this section, no changes to the model are suggested.

##### Changing as a Person

In the original model, this subcategory described the consequences of successful help-seeking in the form of gaining knowledge and positive personal change. Young people in the original model stressed the positive aspects and changes after having experienced mental health problems. Some references were found in the articles with regard to this subcategory, with personal change depicted as finding a more positive outlook on life through one’s own determination and decisiveness.

In this section, the analysed articles provide more aspects on the theme Changing as a person, but no changes to the model are suggested.

Overall, the findings from the analysis aided in developing an elaborated model of help-seeking, Figure 3. The overall notion of help-seeking as a fluid and dynamic process with the three categories Drifting, Navigating and Docking was reinforced.

## 4. Discussion

This scoping review aimed to map published literature on young people’s experiences of seeking help for mental health problems, and to validate whether the previously published model Lost in Space was adaptable as a theoretical model of the help-seeking process. A high level of conformity was found between help-seeking as depicted by the original Lost in Space model and the analysed articles of this study. The analysis reinforced that help-seeking is to be regarded as a fluid and psychosocial process, often experienced by users as unfamiliar and obstacle-laden, tainted by feelings of powerlessness [21,22,23,24].

### 4.1. Discussion and Implications in Relation to the Original Model Lost in Space

After reviewing up-to-date literature on user perspectives of help-seeking for mental health problems among young people, it is clear that the depiction of the initial stage of help-seeking, as being characterized by a sense of drifting, was, to a large extent, corroborated from young people’s experiences described in the reviewed articles. Regardless of context, young people expressed a general feeling of unfamiliarity and a lack of knowledge, often coupled with a sense of insecurity, and trivialisation of experiences [21,22,23,25,26,27,28,29,30]. This was also supported by a large number of codes and expressions relating to the endeavouring strategies theme in an effort to be self-reliant [22,23,25,26,27,28,29,30,31]. This points to the more-or-less continual and relentless nature of the efforts of young people to deal with their problems. A strong wish for self-reliance was consistently stressed in the reviewed articles, with a large variation and number of strategies used to implement self-reliance and deal with mental health problems. Incorporating an elaboration regarding the characteristics of strategies—whether positive/negative, destructive/constructive or approach/avoidant strategies—would provide an additional perspective on how mental health is dealt with by young people. The reviewed articles confirmed that reaching a decision to seek help often takes place with the aid of others [21,22,23,24,25,29,31,32] and distinction between ‘controllers’ and ‘supporters’ in this regard may further elaborate the model. Re-naming of the subcategory Reaching a point of no return into Transitioning towards decision would reflect the transitional nature of the mental health help-seeking process. 

The category Navigating, capturing both personal experiences and structural barriers, was well confirmed by the review. Reflections of not being met by professionals in an appropriate and timely manner, and observations of a perceived lack of resources making access difficult, surfaced in both the original model and the included articles [23,26,29]. Accounts of not being taken seriously, being treated like a child, not listened to and disregarded, indicating power-issues relating to the experiences of young people, as well as descriptions of inconsistent use of support, repeatedly initiating and discontinuing help, appeared in the original model as well as the included articles [22,26,28,29]. Stigma and Trust surfaced as new themes, and Stigma in particular appeared with a large number of codes in the analysed articles. Young people described a feeling of shame, embarrassment, thoughts of what others would think and say and various efforts to conceal both mental health problems and their help-seeking [21,22,23,25,26,27,28,31]. Several articles dealt exclusively with young men and boys, proposing that the reasons for not seeking help were strongly conditioned by gender, with masculine ideals of strength and autonomy acting as obstacles for help-seeking [21,23,28,31]. Similar findings emerged in the Lost in Space model, where issues of self-reliance, wanting to be strong, and shunning displays of weakness, were shared between participants, and were not gender specific. Cultural variations may account for this difference between studies and findings. This said, most participants in studies on help-seeking are female and the findings may translate poorly to other populations and contexts. Help-seeking is exceptionally low among boys and young men, which in itself calls for a focus on specific populations with particularly low help-seeking [33,34].

The latter part of the original model, Docking, was not as well corroborated through the analysis. There were few descriptions of actually finding support and even fewer of personal reflections on the effects of finding help [21,22,31]. It may be that research on the help-seeking process does not focus on support and is discontinued as soon as support is established, and aspects of this may be found in other literature on service utilization or treatment satisfaction. However, by dividing the help-seeking journey into smaller isolated fractions, focus on the process as a whole could be missed, resulting in a stunted model and less understanding of the help-seeking process. Overall, the included articles reinforced the model of help-seeking as a dynamic and psychosocial process, consisting of different stages but without being sequentially fixed.

### 4.2. Discussion and Implications in Practice

This review on help-seeking for common mental health problems included young people from the age of 11 to 25, thus also including young adults. The studies described in the included articles were based on varied recruitment strategies from different contexts. No specific patterns according to age or context could be discerned. The concerns voiced in the studies included themes on structural barriers of the support system, an unfamiliarity and lack of knowledge of mental health and the support system, and simultaneously, a wish for self-reliance, suggesting possible strategies for meeting the help-seeking needs of young people. Although this study aimed to include articles focusing on groups that were wide enough to be defined as population-based, the focus of the included articles tended to be on particular populations, stressing the vulnerability and poor help-seeking of one particular group. Thus, the research had ethno-centric tendencies, whereas there were large overlaps and resemblances of experiences by young people in the help-seeking process regardless of contexts. The attribution of non-help-seeking to stigma and cultural norms amongst Black, Latino and Chinese American youth was observed by others, pointing to this being a more general, rather than group-specific phenomenon [32].

Structural factors, and how young people experience the support system, play an important role in the help-seeking process. Despite different contexts, young people expressed similar concerns relating to issues of availability and accessibility. There were views that waiting times were too long, resources were too few, and in some contexts, that costs and distances posed problems [21,23,25,26,28,29]. Other research has shown that there is a perceived inaccessibility of the support system across different groups of young people regarding resources, entry requirements and coordination between services [13]. Structural obstacles stretch over different geographical and socio-economic backgrounds at the macro level, with high-income countries still showing substantial delays and poor help-seeking rates for young people [14,35]. Thus, even in favourable circumstances, young people perceive structural barriers, pointing to how the support system does not accommodate the fluid and changeable nature of help-seeking. Young people regard mental health as a complex social and relational matter [36]. They often present with diagnostically confusing symptoms, and support systems that are traditionally organized according to medical specialities may not meet the needs of young people with common mental health problems [37]. Integrated youth centres, focusing on meeting young people’s needs in one place through multidisciplinary support with consideration of the context, show promising results [38,39]. In comparison to traditional support, which is by definition siloed and often entails entry requirements according to diagnostic thresholds, integrated youth-friendly services seem to increase help-seeking and access to support, even among groups that are usually hard to reach [33,39]. Studies in a Swedish context have pointed out that youth health clinics providing services to build upon with multi-professional teams and expertise on mental health are available throughout Sweden [40].

Young people reported a lack of knowledge on mental health and the support system, leading to a sense of insecurity and possibly a delay of help-seeking. Improved health literacy among young people may facilitate help-seeking through mechanisms of awareness of service availability and symptom recognition [13]. However, improved help-seeking and mental health among young people may require more than only improved knowledge. Previous reviews have, for example, shown past positive experiences and outcomes of help-seeking and positive contacts with support professionals to be facilitators for seeking help [4,8]. At the same time, a preference for self-reliance when facing mental health problems is consistently reported, with this being particularly prominent in studies with participants having previous experience of mental health problems and mental health support, contradicting the findings of past experiences facilitating help-seeking [8,23]. Young women in particular seem to have poor expectations regarding therapeutic outcomes, signalling a lack of trust in professional supporters, with treatment being perceived as impersonal and protocol-driven [23]. The results of this study identified the importance of supporters’ ability to meet young people responsively, using a person-centred approach. Young people felt more comfortable when the supporters did not use medical language and emphasized the importance of using positive and informal terms for improving communication between the young person and the supporter. Other studies have confirmed this finding, underlining the importance of having young staff who are skilled, respectful, welcoming, and allow for participation and shared decision-making [39].

With this review showing how young people experience mental health help-seeking as a psychosocial and fluid process, often with lack of knowledge and a sense of insecurity, prompt consideration of the organization of present support systems is needed. Young people need to be met in a person-centred and flexible manner. Perhaps, this is where the greatest effort is needed, addressing issues of power from the perspectives of young people, improving opportunities for personal self-reliance and personalized support.

## 5. Methodological Considerations

This review has some limitations. The choice of databases and keywords was developed in accordance with an experienced health literature librarian; however, making a choice always entails the risk that some information may have been missed. Other databases and different keywords may have produced different results. The criteria for including articles were that they should deal with the direct perspectives of young people who had experienced mental health problems and/or help-seeking. Whilst excluding those who had no experience of help-seeking (thus all articles dealing with intentions to seeking help only) might have been a clear-cut and easy choice, that would also have meant that we excluded those with experience of mental health problems who had not sought help for various reasons; thus, avoidance is also a perspective that is worth taking into consideration.

We aimed to include studies focusing on groups wide enough to be defined as population-based; nevertheless, these still often utilized an ethnocentric perspective, such as having a particular ethnic descent. This automatically raises the issue of generalizability and transferability. It was evident that studies consistently focused on particular populations, stressing the vulnerability and poor help-seeking of this particular group. However, similar claims kept reappearing, regardless of which particular group was being studied. A noteworthy phenomenon is that all included articles were published in Western countries. This also limits the transferability of the findings, as young people around the world may be situated in significantly different contexts.

In order to limit bias, the work was conducted by alternating methods of individual and joint reviews. However, subjectivity is a relevant issue that the authors of this review could not completely avoid.

## 6. Conclusions and Implication

The field of help-seeking among young people for mental health problems is receiving growing attention in research and academic literature. However, this review shows that there is substantial heterogeneity among studies with regard to methods, populations and how help-seeking is investigated. In qualitative literature exploring user perspectives, help-seeking is depicted as a fluid, dynamic and psychosocial process, validating the theoretical model of Lost in Space. Important findings include the presence of stigma, a lack of knowledge of mental health issues, a longing for self-reliance and a sense of powerlessness expressed by young people in various contexts and countries. Paying attention to these findings would imply acknowledging young people’s sense of feeling lost, making support services more flexible and person-centred.

## Figures and Tables

**Figure 1 ijerph-19-01430-f001:**
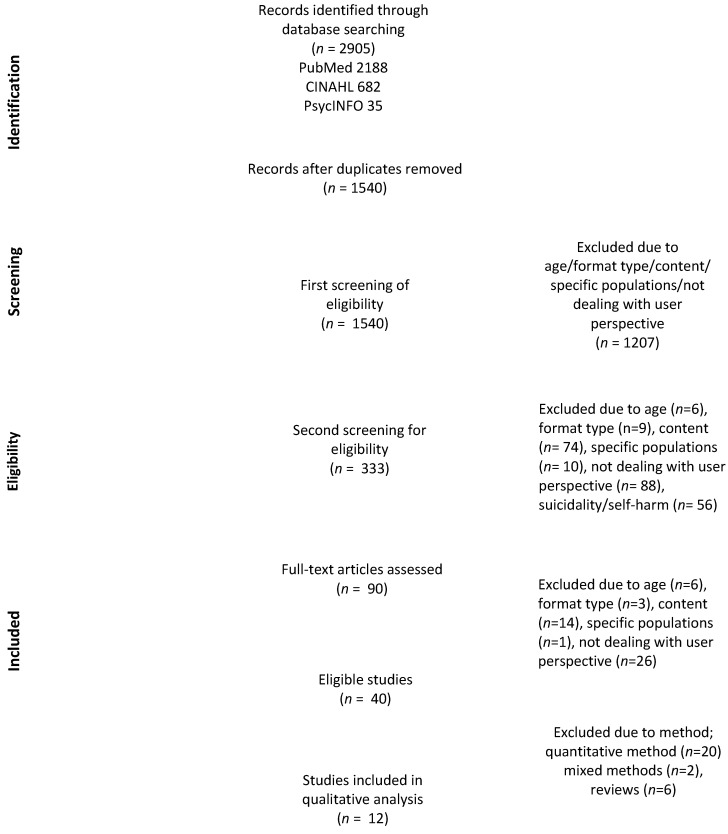
Article search and selection process—PRISMA diagram.

**Figure 2 ijerph-19-01430-f002:**
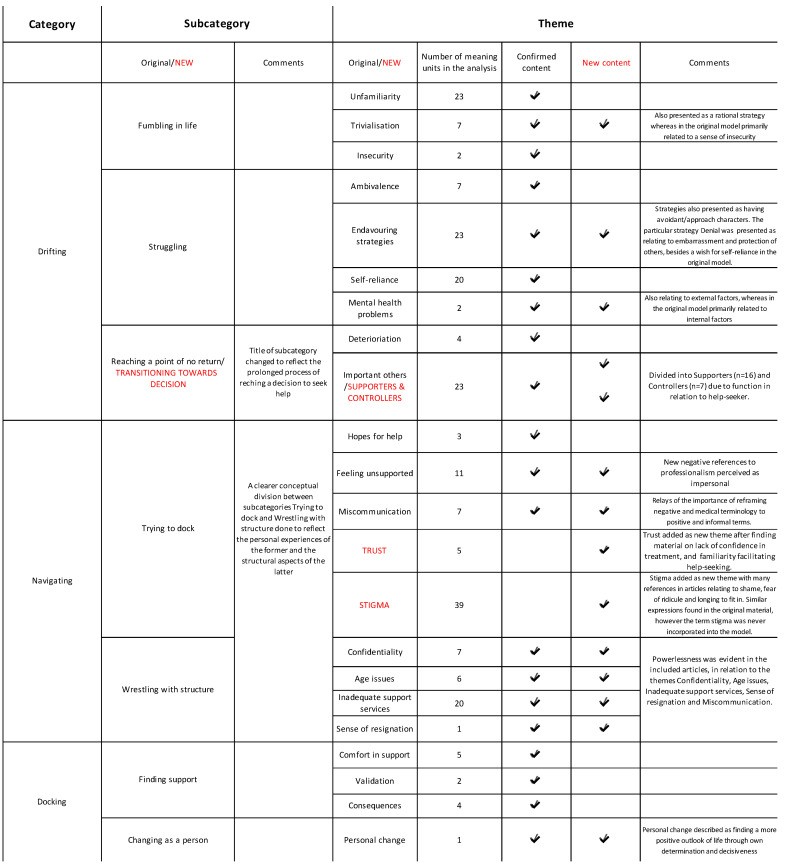
Examination of the help-seeking process from young people’s perspectives.

**Figure 3 ijerph-19-01430-f003:**
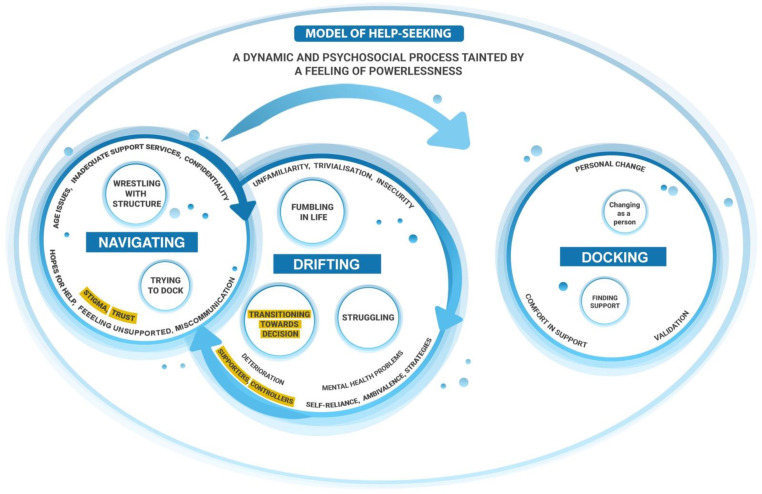
Elaborated and further developed theoretical model of help-seeking among young people for mental health problems.

## Data Availability

Documentation on the database searches, the stepped screening process and the thematic analysis are available from the corresponding author upon reasonable request.

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
