# Peer review of "Mental Health Problems among Young People—A Scoping Review of Help-Seeking"

_ijerph, 2022, doi:10.3390/ijerph19031430_

Round 1
Reviewer 1 Report
Dear Authors!
This valuable contribution is generally well-written and clear. However, it leaves the reader with some questions to which no answer is received.
There are some points I would suggest to address in order to clearify the content.
- How was the thematic analysis performed? More specifically, which codes were identified from the journal articles' texts? (lines 141-142) And particulary, what is the research question referred to in the same pace (lines 141-142) other than finding evidence to support the theoretical model Lost in space? Please insert some sentences regarding this issue. The research question should be clearly formulated in the introduction already.
- Table 1 is the essence of the analysis, therefore it should be somewhat more clear. For instance, what do the words confirmed content - new content refer to?
- The concepts "lost in space" and "navigating" suggest a sort of online help seeking, which is not the case (or it was only for one of the journal articles included in the analysis). Please address this issue clearly and briefly in the introduction.
- The analysis yielded evidence for the theoretical model. It remains, however, a question whether the references relate to only some single themes of the model, or they reinforce the dynamism, that is, the order of stages, too? Please briefly point this out in the analysis, preferably following line 423.
- There are two topics that were not discussed within the article, so it is recommended not to introduce these new issues as the unequal power structures at the very end of the Discussions section (line 480). In the same way, in line 488 the ethnocentric perspective turns up, which had not been in the focus of attention before. However, as these are indeed important issues, I recommend mentioning them in the Discussion part, so that it would not be unexpected and unrelated to the findings.
- None of the articles included in the analysis come from Asia or Africa. The predominantly Western discourse of help seeking for mental health problems seems to be an issue of developed countries. Why so? The theoretical model stems from the Swedish (Western) reality, does it have anything to do with the lack of this topic in non-western academic discourse? Please refer to this issue at the appropriate pace. I wish the authors good luck with the paper and the future work. Reviewer
Author Response
Thank you so much for reviewing our manuscript and providing valuable feedback. Please see below for responses.
1. We have tried to clarify this by mentioning that all themes from the original model were found through coding the analysed articles. Codes from the new material that did not match the original theoretical model subcategories, contributed with new aspects to existing themes of the model and in some cases, generated new themes. This has been added in the Methods section, stage 5. To clarify further, examples of coding was added. The unclear references to research question have been reformulated (in Methods, stage 5).
2. Thank you for your feedback. We have added an explanation in the text of table 1, outlining that ‘original’ refers to subcategories and themes from the Lost in Space model, where findings were corroborated by codes from the analysed articles (confirmed content), other elements that emerged showed further dimensions of experiences that contribute to new perspectives of established subcategories in the model (new content) and that some themes that emerged in the analysis were not readily encompassed within the subcategories in the original model (new) (in Results, 3.2).
3. We have clarified that the help-seeking process is complex and needs increased understanding, whether help-seeking takes place online or in person (in Introduction).
4. Thank you for helping us clarify this. It has been added that the included articles reinforced the model of help-seeking as a dynamic and psychosocial process, where help-seeking consists of different stages but without being sequentially fixed (last in section 4.1).
5. Thank you for pointing this out. The issue of power relates to young people experiencing being treated like a child, not being taken seriously, not being listened to or disregarded. The term ’power’ has been added to the text where such experiences of young people were relayed, hopefully familiarising the reader with this issue earlier, (in section 3.2.2.1, section 3.2.2.2 and section 4.1). The issue of ethno-centrism has been raised in the Discussion-section according to your suggestion (section 4.2).
6. Thank you for this feedback. You are absolutely right that this topic and research area as it stands today, refers predominantly to the Western hemisphere. The UN and the WHO have pointed out that the lack of mental health services in less developed countries is severe, and thus probably accounts for a much smaller academic focus on this area. The theoretical model was based on qualitative work that took place in Sweden, so transferability to highly dissimilar conditions and contexts is unlikely. This has been added in section 5.

Reviewer 2 Report
An interesting read and well written account. A few minor points: Methods - Please indicate whether a scoping review protocol was developed - Please include examples of search terms used or add search strategy as an appendix detailing date of search and number of articles retrieved from each database - Specify the Endnote version - More details could be provided on the Raholm abductive approach to qualitative data thematic analysisAuthor Response
Thank you for reviewing our manuscript and providing valuable feedback. Please see below for responses.
We have changed the wording to clarify that a scoping review protocol was developed and used for undertaking the research (section 2, first paragraph).
Table S1 (supplementary material) details the search strategy.
We have added that EndNote version 20.1 was used (section 2, stage 3).
Thank you for pointing this out. We have added more information on abductive approach (section 2, stage 5).

Reviewer 3 Report
The paper entitled Seeking help for mental health problems among young people – A scoping review is very interesting, actual and relevant for modern readers. Paper presents a scoping review on the topic that is well presented. The aim of the research was to investigate published literature on young people’s perspectives of seeking help for mental health problems, validating the previously developed theoretical model Lost in space. The aim is accomplished.
Introduction to the research is appropriate and informative.
Research is based on good, elaborated and methodologically sound design. Methodology is presented with sufficient details for a reader to imagine the whole research process. Results presented in the paper are significant for contemporary health science. The authors appropriately and objectively interpreted and discussed presented results. Conclusions are sound. The paper presents results that can be considered as relevant in the field. Strengths: good methodological base, development of theoretical model, objectivity, research caution (awareness of limitations), data systematization, awareness on the importance of the topic, analytic and synthetic approach to the research topic. I enjoyed reading the paper and I recommend the paper for publishing.
Author Response
Thank you so much for your expert opinion of our manuscript.